# A Clinical Trial to Evaluate the Efficacy and Safety of 3D Printed Bioceramic Implants for the Reconstruction of Zygomatic Bone Defects

**DOI:** 10.3390/ma13204515

**Published:** 2020-10-12

**Authors:** Ui-Lyong Lee, Jun-Young Lim, Sung-Nam Park, Byoung-Hun Choi, Hyun Kang, Won-Cheul Choi

**Affiliations:** 1Department of Oral and Maxillofacial Surgery, Chung-Ang University Hospital, Soeul 06973, Korea; 2Chung-Ang 3D Craniofacial Research Society, Chun-Ang University, Seoul 06974, Korea; 3CGbio 3D Innovation Center, Seongnam-si 13211, Korea; jylim77@cgbio.co.kr (J.-Y.L.); snsimin@daewoong.co.kr (S.-N.P.); byounghunchoi@daewoong.co.kr (B.-H.C.); 4Department of Anesthesiology and Pain Medicine, Chung-Ang University College of Medicine, Seoul 06974, Korea; roman00@naver.com; 5Department of Orthodontics, Chung-Ang University Hospital, Seoul 06974, Korea

**Keywords:** bioceramic, patient-specific, 3D printing, additive manufacturing, clinical trial

## Abstract

The purpose of this study was to evaluate the clinical efficacy and safety of patient-specific additive-manufactured CaOSiO_2_-P_2_O_5_-B_2_O_3_ glass-ceramic (BGS-7) implants for reconstructing zygomatic bone defects at a 6-month follow-up. A prospective, single-arm, single-center, clinical trial was performed on patients with obvious zygoma defects who needed and wanted reconstruction. The primary outcome variable was a bone fusion between the implant and the bone evaluated by computed tomography (CT) at 6 months post surgery. Secondary outcomes, including implant immobilization, satisfaction assessment, osteolysis, subsidence of the BGS-7 implant, and safety, were assessed. A total of eight patients were enrolled in the study. Two patients underwent simultaneous reconstruction of the left and right malar defects using a BGS-7 3D printed implant. Cone beam CT analysis showed that bone fusion at 6 months after surgery was 100%. We observed that the average fusion rate was 76.97%. Osteolysis around 3D printed BGS-7 implants was not observed. The mean distance displacement of all 10 implants was 0.4149 mm. Our study showed no adverse event in any of the cases. The visual analog scale score for satisfaction was 9. All patients who enrolled in this trial were aesthetically and functionally satisfied with the surgical results. In conclusion, this study demonstrates the safety and promising value of patient-specific 3D printed BGS-7 implants as a novel facial bone reconstruction method.

## 1. Introduction

Tumor resection, trauma, osteoradionecrosis, and various other causes may result in maxillofacial bone defects. The ultimate goal of reconstruction of maxillofacial bone defects is to restore bone defects and facial shape to their original form, minimize malocclusion, and restore masticatory function. In addition, the morbidity of the donor and recipient should be minimized and reconstruction should be done with a minimum number of operations [1]. There are many options for the reconstruction of maxillofacial bone defects. Grafting with allograft bone has been documented to be a useful tool in reconstructing jawbone defects [2,3]. Among them, microvascular bone transfer has become the gold standard for reconstruction. However, donor site morbidity, prolonged hospitalization time, and a lengthy recovery period with delayed functional rehabilitation are frequently noted [4]. Hence, there is an increasing need for an alternative option to microvascular free osteocutaneous flaps [5]. Ready-made implants made of various materials are also used to reconstruct maxillofacial defects. However, ready-made implants are inconvenient because they require secondary processing before implantation in accordance with the anatomical structure of the patient. In order to overcome this, patient-specific implants using 3D printing technology have recently been developed. Since the implants developed by 3D printing technology are customized, secondary processing is not necessary, and the implants can be transplanted quickly.

The development of CAD–CAM (computer aided design/computer aided manufacturing) technology and additive manufacturing with adequate strength and biocompatibility makes the foundation for the application of patient-specific three-dimensional printed titanium implants in the field of maxillofacial bone reconstruction [6,7,8]. A multicenter study using patient-specific CAD/CAM reconstruction plates for mandibular reconstruction has been performed, showing good outcomes [9], while a case was reported about the successful reconstruction of facial bone defects using a 3D printed titanium mesh [5]. Recent success in the manufacture of patient-specific 3D printed metal devices by additive manufacturing has been demonstrated [1,5,10].

However, titanium implants can eventually bring about subsidence because of the stress shielding effect and the differences in the modulus of elasticity [11]. Moreover, metallic artifacts due to the titanium pose difficulty in the evaluation of bone–implant fusion and detection of possible pathology in the maxillofacial area. To overcome these limitations, bioactive glass-ceramics with improved mechanical strength have been developed for bone grafting and repair. These bioactive glass-ceramics form apatite layers in the physiological condition of the bone and chemically bind to bone directly, resulting in improved bone-bonding strength [12,13].

Among the many types of bioactive glass-ceramics, CaOSiO_2_-P_2_O_5_-B_2_O_3_ glass-ceramics (BGS-7) have been reported to induce osteoblastic differentiation of human mesenchymal stem cells, which results in an improved bone–implant contact ratio [14,15,16]. Furthermore, the intravenous administration of BGS-7 in rats did not show any toxicity for 90 days [17]. A 4-year follow-up study demonstrated similar fusion rates and clinical outcomes in both BGS-7 spine cage and autologous bone with a titanium cage in one-level posterior lumbar interbody fusion [11]. Besides, technology has been developed to make BGS-7 into a patient-specific implant through a fused deposition manufacturing method.

This clinical trial was designed to evaluate the safety and effectiveness of the patient-specific implant, manufactured by 3D printing process technology with BGS-7 for zygoma reconstruction.

## 2. Materials and Methods

### 2.1. Study Design

The Investigational New Drug application (IND) was approved by the Korean Food and Drug Administration (761) for this clinical trial. IND means that the clinician who intends to conduct clinical trials using the drug or medical device to collect safety efficacy data for the human body applies for approval from the Korean Ministry of Food and Drug Safety. The study protocol was approved by the Ethics Committee of Chung-Ang University Hospital (IRB no.3DB-1701) and registered at cris.nih.go.kr (KTC0002655). This study was performed according to the principles of the Declaration of Helsinki, 2000 [18]. In addition, written informed consent was obtained from all participants before participating in the trial. The study has been registered in the Korean Trial Register (NTR1289). The primary outcome variable was the fusion between the implant and the bone evaluated by computed tomography (CT) at 6 months after surgery. Secondary variables, including immobilization of the implant by CT, a visual analog scale (VAS) for satisfaction, osteolysis, subsidence of the BGS-7 implant, and safety, were assessed. A 4-year follow-up study evaluating fusion rates and clinical outcomes using BGS-7 spine cage in 1-level posterior lumbar interbody fusion has been performed [9]. Since the spine study used the BGS-7 spine cage, which is the same material used in this study, the primary outcome and secondary outcomes used in that study were also evaluated in this study.

The inclusion criteria and exclusion criteria of this study were as follows (Table 1).

### 2.2. Computer-Aided Virtual Surgery, Design, and Fabrication of Patient-Specific Devices

Cone beam computed tomography (CBCT, 3D eXam, Kavo Dental GmbH, Biberach, Germany) of the patients was taken while patients were in an upright position. The CBCT data of the maxillofacial regions were obtained with a 0.4 mm voxel size and 512 × 512 matrices using 120 kVp, 11 mA, 17.8 s of scan time, and a 12-inch detector field (Figure 1A). Patient data were stored in the Digital Imaging and Communications in Medicine (DICOM) format and reconstructed into 3D bone images using the Mimics program (Materialise Co, Leuven, Belgium). By using the thresholding function of the program, we perform the process of mask-drawing the bone part for each slice cut, and 3D rendering by connecting each mask through the 3D calculation function. Following the completion of the 3D bone image-rendering, zygoma implants were designed using 3 Matics software (Materialise Co, Leuven, Belgium). The zygomatic bone defect is to be reconstructed accurately and matched to the original defect of the patient. The defect part is designed to be smoothly connected based on the normal bone, but the shape may be designed slightly differently depending on the surgical method or the fixing method selected by the surgeon. The basic contour of the reconstruction implant was taken from the other side of the non-defect zygomatic bone. The design can be modified according to the request of the surgeon.

After loading the stereolithography (STL) file of the implant with the confirmed design, we set 3D printing conditions such as printing position, printing direction, resolution, nozzle size, and support configuration. During 3D printing, the stacking direction is set parallel to the force direction that the implant will receive when the implant is later inserted into the body. In consideration of the occurrence of shrinkage during the sintering process, ceramics must be built up to a larger size than the final sample. BGS 7 powder was mixed with the binder according to the mixing ratio using a mixer to make the paste. This paste was filled in a stainless-steel syringe barrel and installed on the printer. After installation, printing conditions such as speed and discharge amount were appropriately set, and additive manufacturing was performed using the fused deposition modeling method (video). Additive-manufactured products were dried overnight.

The dried additive-manufactured products were placed in a furnace, and then, a sintering process was performed. As the heating process proceeded, the strength of the product increased, and shrinkage occurred. After the sintering was completed, the product was washed and dried using an ultrasonic cleaner. Cleansing was performed to remove any foreign substances or residues that may remain after removal of the support. After cleansing and drying, the product was put in a wrapping paper, and the packaging paper was hot-sealed, and then, gamma-ray sterilization was performed at a dose of 15–25 kGy.

### 2.3. Surgical Procedure

All cases were reconstructed with a bioceramic 3D printed zygoma implant (EASYMADE, CG Bio Co, Seoul, Korea), and implants were fixed using a mini-plate or wire via a transoral approach. Intraoperatively, the fibrous tissue filled the gap of nonunion. The soft tissue interposed between the non-united bones has to be removed completely. After removing all the fibrous tissue and exposing the bony margin of both sides, the 3D printed implant was applied and fixed with a mini-plate and or wire (Figure 1C). The method of fixing a screw hole through the implant itself was considered. However, since there was a possibility that the implant could break during the screw tightening process, the plate was fixed by pressing the implant downward (Figure 1D). Therefore, depression was designed to be placed on the implant (Figure 1B). Alternatively, we drilled a small hole in the implant and fixed it using a wire through the hole. In some cases, plates and wires were used together to fix them. All patients received the same anesthetic protocol. Anesthesia was induced with intravenous administration of 2 mg/kg propofol and 0.6 mg/kg rocuronium. Using 2–3% sevoflurane in 1.5 L/min N_2_O and 1.5 L/min O_2_, anesthesia was maintained. Noninvasive arterial blood pressure (NIBP), electrocardiography, and pulse oximetry were monitored continuously during the operation. Lactated Ringer’s solution (3–6 mL/kg/h) and 6% hydroxyethyl starch were infused throughout surgery. A total of 30 mg ketorolac was administered just before the end of surgery for pain control and 5 mL of dexamethasone was administered intravenously after surgery.

### 2.4. Evaluation

The primary outcome variable was the fusion between the implant and the bone evaluated by computed tomography (CT) at 6 months after surgery. The other variables assessed included immobilization of the implant by CT, a visual analog scale (VAS) for satisfaction, osteolysis, subsidence of the BGS-7 implant, and safety.

The VAS is a 10 cm long horizontal line. The patient expressed his or her satisfaction by pointing a vertical mark on the 10 cm long line. The measurement in centimeters was changed to the same number of points, ranging from 0 to 10 points. The question was, “Are you satisfied with your zygoma implant?”

The evaluation of CT used for bone fusion was performed by an independent and experienced oral and maxillofacial surgeon who did not participate in the clinical trial. Surgical variables such as operating time, blood loss, length of hospital stay in addition to adverse events, and post operative complications were also documented.

### 2.5. Cone Beam CT and 3D Comparison

CBCT was performed immediately after surgery and 6 months after surgery to evaluate bone fusion, the fusion area, and the immobilization of implants. A CT Digital Imaging and Communications in Medicine (DICOM) file, taken 6 months after surgery, was loaded into the in vivo 5 software (Anatomage, San Jose, CA, USA). The multi-planar plane was re-oriented so that the longest axis of the implant was perpendicular to the reference plane, and the minor axis was horizontal to the reference plane (Figure 2B–D). A cross-section was created based on the newly reorientated plane (Figure 2A).

To assess bone fusion, the newly formed axial planes (Figure 2A) were evaluated one by one at 0.4 mm intervals. “Fusion” is defined as the case where complete bone contact occurs in more than 50% of the total area of the interface where the implant meets the bone. Each section was checked, and the total number of sections was counted. For each case, we calculated the ratio of the number of fused cross-sections to the total number of cross-sections. Osteolysis around the BGS-7 implants and subsidence to the surrounding bone were both measured using axial, sagittal, and coronal cuts. An oral and maxillofacial surgeon independently reviewed and scored them as “fusion”, if the calculated fusion rate was over 50%.

CT data taken immediately post surgery (reference data) and 6 months after surgery (measured data) were converted to stereolithography (STL) files and imported into Geomagic Control X software (3D systems, Rock Hill, SC, USA), and initial alignment was applied to superimpose the measured data onto the reference data (Figure 3A). A best-fit algorithm was implemented to overlay the measured data taken to the reference data, and the 3D Compare tool was utilized to measure deviation at 10 evenly spaced points on the implant surface (Figure 3B).

### 2.6. Sample Size Determination

This study was a prospective, single-arm, single-center, clinical trial aimed at collecting initial safety and efficacy information for medical devices, designing follow-up clinical trials, providing criteria for evaluation items, and evaluation methods. Because it is an exploratory clinical trial, it does not statistically calculate the number of subjects.

### 2.7. Statistical Analysis

All data were analyzed using SPSS (Statistical Package for the Social Sciences) (IBM, Chicago, IL, USA) (α = 0.05). Between-case differences for distances between random points were analyzed with a linear mixed-effects model (LMEM), which was made with cases as independent fixed factors, and each distance as a random effect. The number of cuts was analyzed with an LMEM, which was made with cases and fusion status as independent fixed factors, and each cut as a random effect.

## 3. Results

### 3.1. Baseline Characteristics

Between April 2017 and January 2019, a total of eight patients were enrolled in the study. Two patients underwent simultaneous reconstruction of the left and right malar defects using a BGS-7 3D printed implant (Cases 4, 5, 9, and 10). One patient (Case 3) refused to continue in the study 3 months after surgery. Consequently, the study was performed with Case 3’s CBCT taken at 3 months after reconstruction. Of the 10 cases, one case was a male (10%). The mean age was 34.8 years, and none of the patients were smokers. All patients presented with various sizes of zygomatic bone defects in various areas due to previous malarplasty. Baseline characteristics are shown in Table 2.

### 3.2. Fusion Rate Analysis Using 3D CT

Cone beam CT analysis showed that all patients exhibited bone fusion at 6 months after surgery (10/10). According to the results of analyzing the fusion rate using CBCT, we observed that the average fusion rate was 76.97%, and the standard deviation was 11.36. In all cases, the fusion rate exceeded 50%. The minimum was 58.33%, and the maximum was 88.24% (Figure 4). The LMEM showed no evidence of significant differences in average fusion rate between each cases (F(9, 9) = 0.900, *p* = 0.561), but significant differences for fusion status were observed (F(1, 9) = 40.438, *p* < 0.001) (Figure 4). In this study, osteolysis around the 3D printed BGS-7 implants was not observed.

### 3.3. 3D Comparison between Immediate Post Operative and 6 Months Post Operative

Table 3. shows the distance displacement of the measurements at 10 randomly distributed points on the 3D printed BGS-7 implants. There were statistically significant differences in the distance displacement of the 10 BGS-7 implants (*p* < 0.001). LMEM also showed a significant difference between cases (F(9, 16.76) = 6.095, *p =* 0.001) (Figure 5). The maximum and minimum mean distance displacements were 0.889 and 0.1156 mm, respectively, while the mean deviation distance displacement of all 10 implants was 0.4149 mm.

In the color map using the 3D inspection program, the displacement was the same as the measured displacement (Figure 6). In Figure 6, if the outer surface is blue, it means that the CT taken at 6 months after surgery is lower than the CT taken immediately after surgery. Most implants were displaced downward 6 months after surgery when compared to CT taken immediately after surgery. This result shows that the implant is attached closer to the surrounding bone under the downward pressure of the skin.

### 3.4. Results of Satisfactory Scale and Adverse Events

The VAS (Visual Analogue Scale) score for satisfaction was 9 at 6 months after surgery. All patients who performed the survey were aesthetically and functionally satisfied with the results of the operation. In case 3, we were unable to measure VAS because of loss to follow-up 3 months after surgery. In this follow-up study, there were no post operative complications, and no adverse events were reported in any of the patients.

## 4. Discussion

Reduction malarplasty is frequently utilized to contour the facial bone in Asia by repositioning the prominent malar bone. Improper fixation can result in nonunion or malunion of the malar complex due to the pulling force of the masseter muscle [19]. Consequently, the malar complex is displaced inferiorly and rotated externally, which leads to facial deformity, cheek drooping, or jaw movement restriction [20].

Instead of repositioning the displaced malar complex, which is due to malunion or nonunion after zygoma plasty, reconstruction of the bony gap in the zygomatic bone with a prosthesis can be a good alternative. Reconstruction surgery using a prosthesis is much simpler and less time consuming than repositioning the displaced malar complex and can be performed using a transoral approach. Most patients complain of the depression that occurred on the skin covering the gaps in the zygoma. They wanted to reduce this depression. After reconstruction surgery, all patients were satisfied with the aesthetics because the dents on their faces disappeared.

The development of implant materials has evolved rapidly over the years. In addition, the development of CAD–CAM technology and titanium 3D printing with adequate strength and biocompatibility has led to the application of patient-specific titanium implants in maxillofacial bone reconstruction [6,7,8]. However, titanium implants can eventually lead to subsidence due to the difference in the stress shielding effect and elastic modulus [11]. In addition, titanium metal artifacts can make bone assessments, such as fusion with implants and detection of possible pathologies in the maxillofacial region, difficult. To overcome these limitations, patient-specific additive-manufactured implants made of BG (BGS-7) were tested through this clinical trial. Among the many types of bioactive glass-ceramics, CaOSiO_2_-P_2_O_5_-B_2_O_3_ glass-ceramics (BGS-7) have been reported to induce osteoblastic differentiation of human mesenchymal stem cells, which results in an improved bone–implant contact ratio [14,15,16]. Previous in vitro and in vivo studies have demonstrated that the BGS-7 cage for implantation in posterior lumbar interbody fusion has high bioactivity and chemical bonding ability [11,14,15,16]. The BGS-7 interbody fusion cages used for spinal surgery are not patient-specific and ready-made. The shape of the spine cage is not complicated and has a large contact area with the bone.

In contrast, implants for reconstructing zygoma defects vary in shape and size, and the contact area with zygomatic bone is not wide. However, since it is not a site under a large load, we conducted a clinical trial on malar defects for the first time to evaluate the efficacy and stability of the 3D printed bioceramic implant. As far as we know, this is the first clinical trial to reconstruct a facial bone using a patient-specific bioceramic implant.

Due to the lack of clinical data on facial bone reconstruction using additive-manufactured bioceramic implants, this trial was designed to evaluate clinical and radiological outcomes of reconstruction with bioceramic 3D printed implants. This clinical trial on patients receiving additive-manufactured bioceramic implants has shown that the implants are safe and result in clinical and radiological improvement. The lack of a control group is a weakness of this study. However, there was no standard treatment for the bony gap on the zygoma that occurred after the zygoma plasty, and the number of patients undergoing zygoma reconstruction surgery was small, so it was almost impossible to make a control group.

The authors evaluated the fusion rates of implants with bone using CBCT. A common method for assessing bone—implant ° integration is to measure bone–implant contact area. In the case of a dental implant, since the dental implant is drilled on a compact bone and the implant is placed, it is theoretically possible for the entire implant surface to make contact with the bone. However, in the case of a zygoma implant, the implant is placed by placing it on the bone. As this implant is designed to reconstruct the gap, there are many areas where the implant and bone do not make contact. Therefore, for these implants, a new standard for measuring the fusion rate is needed. In this study, the fusion rate was calculated by calculating the ratio of the area in contact with the bone to the area designed to contact the bone during implant design. The average fusion rate was 76.97%, and the standard deviation was 11.36. In all cases, the fusion rate exceeded 50%, with the minimum being 58.33%, and the maximum being 88.24%. In this study, osteolysis around 3D printed BGS-7 implants was not observed. Statistical analysis showed no evidence of a significant difference between cases (F(9, 9) = 0.900, *p =* 0.561), but significant differences for fusion status were observed (F(1, 9) = 40.438, *p <* 0.001) (Figure 4).

The mean removal torque of BGS-7 implants installed in rabbit ileum was significantly higher than that of titanium and PEEK (Polyether ether ketone) at 2 and 4 weeks, and the other three implant types at 8 weeks [15]. In vitro study results showed sufficient hydroxycarbonate apatite layer formation on CaOSiO_2_-P_2_O_5_-B_2_O_3_ glass-ceramics (BGS-7) after soaking in simulated body fluid [21]. In addition, CaO-SiO_2_-P_2_O_5_-B_2_O_3_ glass-ceramics are known to be used as a coating material to enhance osseointegration between bones and implants [14,22]. Titanium implants coated with hydroxyapatite have been shown to accelerate bone healing and prompt an enhancement in initial implant osseointegration [23,24,25,26,27,28,29,30]. The method used to fix the implant in this clinical trial is to fix it by pressing the implant using the mini-plate and a wire. Initial stability is important for proper osseointegration in dental implants [19,20,21].

Consequently, it was especially important to evaluate whether the BGS-7 implant was immobilized at the planned position until fusion with the bone. We analyzed the immobilization of the delivered implants using 3D comparison. A best-fit algorithm superimposed CT data taken immediately after surgery and 6 months after surgery, and deviation at 10 evenly spaced locations on the implant surface was measured (Figure 6). The maximum and minimum mean displacements were 0.889 and 0.1155 mm, respectively, while the mean distance displacement of all 10 implants was 0.415 mm. Most implants were displaced downward 6 months after surgery when compared to immediately after surgery. It can be seen that the implant was attached closer to the surrounding bone. Our study showed no adverse events in any of the cases. Therefore, this clinical trial shows that excellent osseointegration of the BGS-7 implants promotes effective bone fusion around the zygomatic bone. The BGS-7 implants can be successfully fixed using a mini-plate or wire. When performing zygoma reconstruction surgery using a patient-specific BGS-7 implant, most of the surgery took less than an hour because there was no need to harvest bones elsewhere in the body. Patients were discharged the day after surgery because there was no donor site morbidity and it took less time for surgery. As hospitalization period was shortened, overall medical expenses also decreased (data not shown).

## 5. Conclusions

The 6 months follow-up showed a high fusion rate and clinical results for reconstructing malar defects using patient-specific 3D printed BGS-7 implants. Furthermore, all patients who enrolled in this trial were aesthetically and functionally satisfied with the results of the operation with BGS-7 implants. Therefore, this study demonstrates the safety and promising value of patient-specific 3D printed BGS-7 implants for new facial bone reconstruction.

## Figures and Tables

**Figure 1 materials-13-04515-f001:**
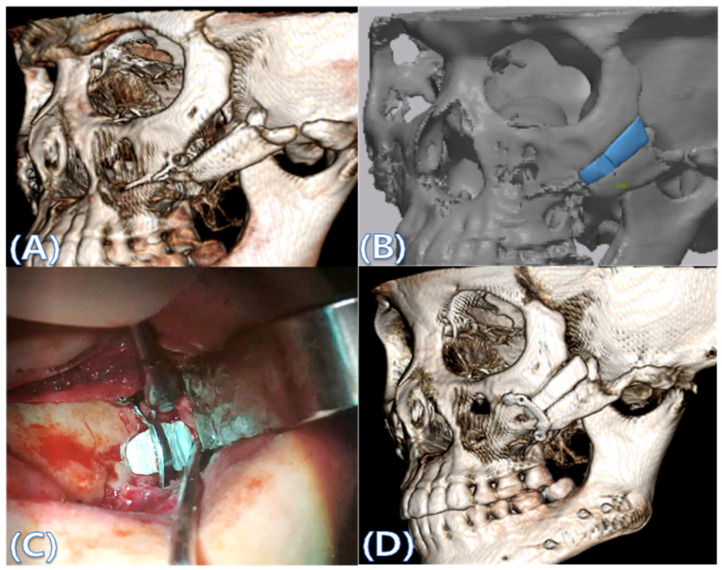
(**A**) The malar complex is dislocated inferiorly and rotated externally, which leads to facial asymmetry, bony gap, and facial drooping. (**B**) The implant for reconstructing the defected part is basically designed to be smoothly connected based on the normal bone. (**C**) After removing all the intervening soft tissue and exposing the bony margin of both sides, a 3D printed implant was applied and fixed with a mini-plate. (**D**) CT taken at 6 months after surgery shows that the bony defect is well reconstructed.

**Figure 2 materials-13-04515-f002:**
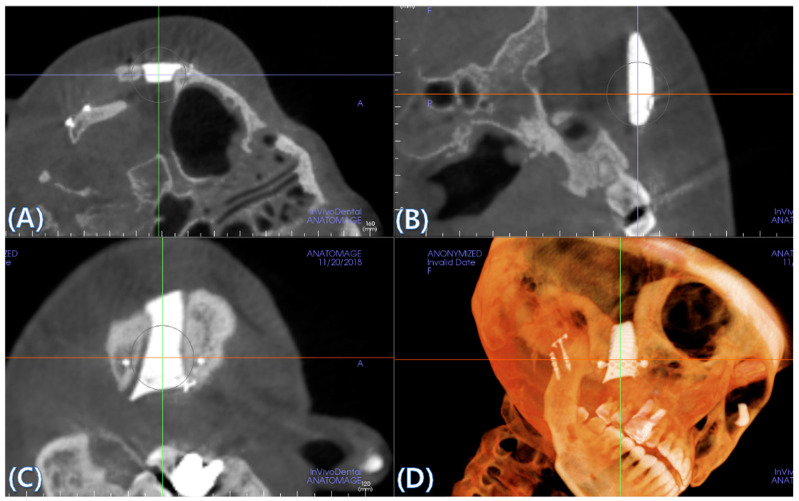
(**A**) A cross-section is created based on the newly reorientated plane. (**B**–**D**) The multi-planar plane is re-oriented so that the longest axis of the implant is perpendicular to the reference plane and the minor axis is horizontal to the reference plane.

**Figure 3 materials-13-04515-f003:**
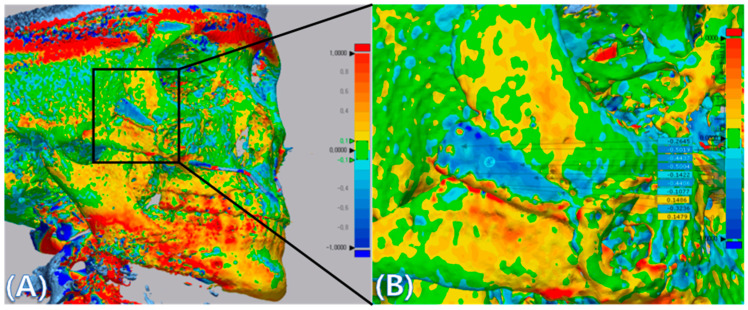
(**A**) CT data were converted into stereolithography (STL) files and imported into Geomagic Control X, and superimposition of CT data taken immediate post surgery (reference data) and CT data taken at 6 months after surgery (measured data) was performed using the Geomagic Control X software. The images were positioned using both the Initial Alignment and Best-Fit Alignment tools. The colors show overlays of 2 CT scan data. (**B**) Deviations between reference data and measured data were measured at ten evenly spaced points on the implants. The specific values of deviation on the implants are demonstrated. The value of deviation across the entire scan is color-graded from 1 mm (blue) to +1 mm (red).

**Figure 4 materials-13-04515-f004:**
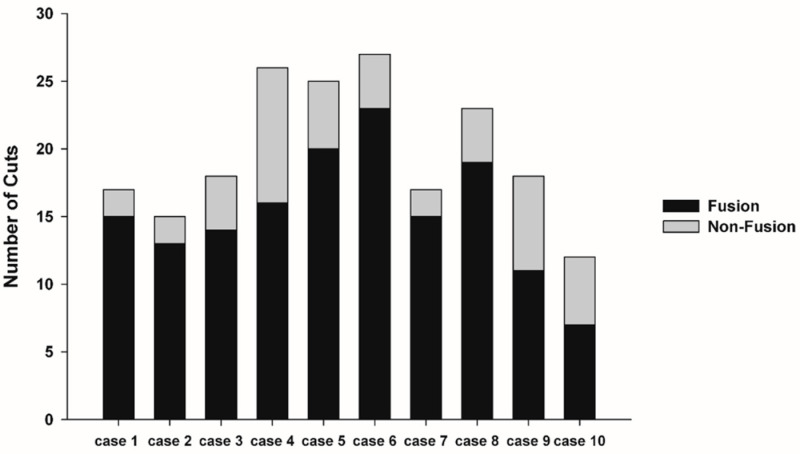
Linear mixed-effects model (LMEM) showed no evidence of significant difference between each cases (F(9, 9) = 0.900, *p =* 0.561), but significant differences for fusion status (F(1, 9) = 40.438, *p <* 0.001).

**Figure 5 materials-13-04515-f005:**
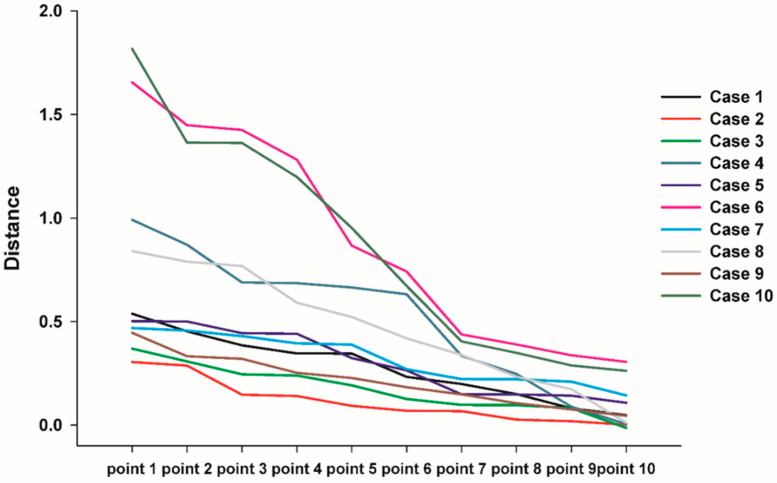
Linear mixed-effects model (LMEM) also showed significant differences between cases (F(9, 16.76) = 6.095, *p =* 0.001).

**Figure 6 materials-13-04515-f006:**
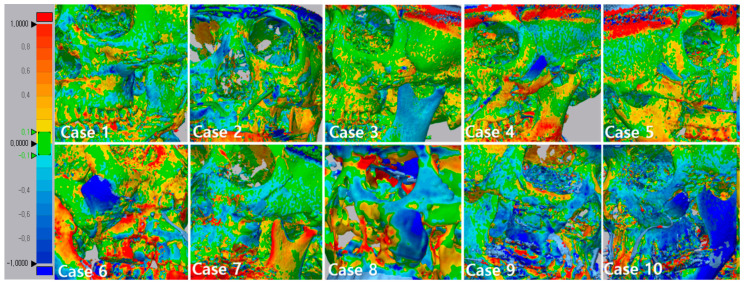
3D comparison between immediately after surgery and after 6 months. This figure shows that the implant was attached closer to the surrounding bone. The specific values of deviation on the implants are demonstrated. The value of deviation across the entire scan is color-graded from 1 mm (blue) to +1 mm (red).

**Table 1 materials-13-04515-t001:** The inclusion criteria and exclusion criteria of this study.

Inclusion Criteria	Exclusion Criteria
1. Adults aged 19 to under 75 who have completed craniofacial growth 2. Those with obvious zygoma defects, who need and want reconstruction 3. In the case of women of childbearing potential, those who consent to contraceptive use during the clinical trial participation period (more than 6 months after surgery) 4. Those who voluntarily agreed to participate in clinical trials and were willing to comply with the study protocol.	1. People with uncontrolled metabolic diseases (e.g., diabetes, osteomalacia, thyroid disease) 2. People who are taking or are planning to take drugs that can affect bone metabolism (Bisphosphonate, Recombinant human parathyroid hormone, Denosumab, etc.) 3.Those with uncontrolled gingivitis, periodontitis, and dental caries 4. People with severe heart disease or severe liver dysfunction 5. Persons with infectious diseases with a risk of recurrence 6. People with blood diseases (leukemia, hemophilia, sepsis, etc.) 7. People who cannot stop taking steroids, anti-thrombotics, or anticoagulants before surgery 8. People who cannot stop systemic corticosteroids or anabolic steroids for 3 months after surgery 9. People with osteomalacia and Paget’s disease 10. People who have experienced radiation therapy at the surgical site 11. People who are allergic to implant materials 12. Patients with syphilis and severe epilepsy 13. Drug abuse or alcoholics 14. Smokers 15. People who are pregnant or have a pregnancy plan during the clinical trial period 16. In case the investigator judges that participation in the clinical trial is inappropriate because other ethical or clinical trial results may be affected.

**Table 2 materials-13-04515-t002:** Summary of demographic characteristics, zygoma defect configuration, and virtual design of a patient-specific implants.

Screening Number	Gender/Age	Cause of Defect	Defect	Defect Area (mm)	Virtual Planning	Fixation Method
1S-01	F/29	Zygoma plasty	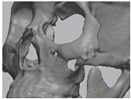	X = 16.4 Y = 9.8 Z = 3.8	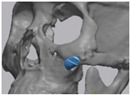	Plate
1S-02	F/24	Zygoma plasty	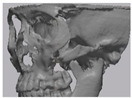	X = 32.4 Y = 12 Z = 4.8	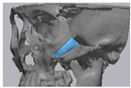	Plate
1S-03	M/25	Zygoma plasty	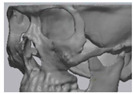	X = 17.3 Y = 13.3 Z = 5	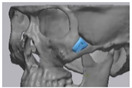	Plate
1S-04	F/28	Zygoma plasty	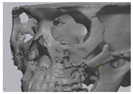	X = 28.4 Y = 10.9 Z = 5	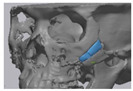	Plate
1S-05	F/28	Zygoma plasty	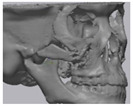	X = 26 Y = 8.2 Z = 4.4	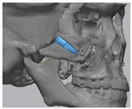	Plate
1S-06	F/41	Zygoma plasty	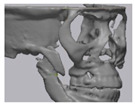	X = 29 Y = 19.2 Z = 5	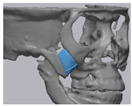	Plate and wire
1S-07	F/26	Zygoma plasty	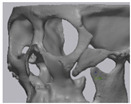	X = 18.1 Y = 7 Z = 4.1	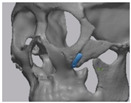	Plate
1S-08	F/53	Zygoma plasty	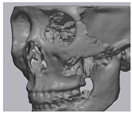	X = 37 Y = 21.5 Z = 5	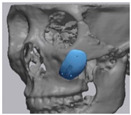	Plate and wire
1S-09	F/53	Zygoma plasty	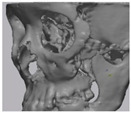	X = 21.8 Y = 13.3 Z = 5	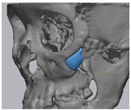	Wire
1S-10	F/53	Zygoma plasty	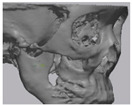	X = 16.8 Y = 13 Z = 4	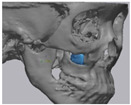	Wire

**Table 3 materials-13-04515-t003:** The displacement of the measurements at 10 randomly distributed points on the 3D printed BGS-7 implants.

	Point 1	2	3	4	5	6	7	8	9	10	Aver.	SD
Case 1	0.3853	0.3461	0.5375	0.3455	0.4526	0.233	0.1499	0.1982	0.0476	0.0812	0.2776	0.1616
2	0.1469	0.3044	0.0014	0.2874	0.0191	0.1403	0.0927	0.0263	0.0695	0.0675	0.1155	0.1064
3	0.0979	0.0137	0.1259	0.0966	0.0821	0.1918	0.2396	0.3076	0.2452	0.3689	0.1741	0.1170
4	0.3342	0.0045	0.0888	0.2472	0.6319	0.6853	0.6889	0.6646	0.871	0.9915	0.5207	0.3327
5	0.1479	0.3236	0.1486	0.1077	0.4406	0.1422	0.5004	0.4437	0.5019	0.2645	0.3021	0.1602
6	0.4378	0.3892	0.337	0.3052	0.8668	0.7413	1.2819	1.4485	1.4251	1.6551	0.8887	0.5231
7	0.4298	0.2216	0.3879	0.2224	0.2695	0.2097	0.4689	0.3944	0.1436	0.456	0.3203	0.1192
8	0.4187	0.3392	0.5911	0.8404	0.5219	0.7681	0.7887	0.2335	0.1735	0.011	0.4686	0.2826
9	0.3199	0.1057	0.2524	0.1483	0.2272	0.3322	0.1833	0.4457	0.0755	0.045	0.2135	0.1270
10	0.3489	0.2618	0.9525	0.4041	0.288	1.3645	1.1981	0.6722	1.8174	1.3628	0.8670	0.5513

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
