# Peer review of "A Clinical Trial to Evaluate the Efficacy and Safety of 3D Printed Bioceramic Implants for the Reconstruction of Zygomatic Bone Defects"

_materials, 2020, doi:10.3390/ma13204515_

Round 1
Reviewer 1 Report
This study, presented as a clinical trial, aims to evaluate the clinical efficacy and safety of specific bioactive glass-ceramic implants for reconstructing zygomatic bone defects. Although this is an area of considerable interest and this study has some strengths, there are several points that need clarification and improvement. In addition, a substantial English language revision and editing work are strongly recommended.
Abstract
1- In the abstract is missing the number of participants enrolled in this trial. Please, add this information.
2- At line 21, it is not clear if “other variables” are considered as secondary outcomes. Please, specify this sentence.
Introduction
1- At line 36, please replace “its” with “their”.
2- Introduction should be slightly improved and elaborated when you assert that “there are many options for the reconstruction of maxillofacial bone defects” (lines 38 - 40).
Please, add some sentences regarding the use of bone allografts in oro-maxillofacial surgical interventions and references such as “Barone, A.; Covani, U. Maxillary alveolar ridge reconstruction with nonvascularized autogenous block bone: clinical results. J Oral Maxillofac Surg 2007 ,65, 2039-46” and “Grassi, F.R.; Grassi, R.; Vivarelli, L.; Dallari, D.; Govoni, M.; Nardi, G.M.; Kalemaj, Z.; Ballini, A. Design Techniques to Optimize the Scaffold Performance: Freeze-dried Bone Custom-made Allografts for Maxillary Alveolar Horizontal Ridge Augmentation. Materials 2020, 13, 1393” may be enclosed.
3- At line 47 there are two missing spaces: “by3D” and “arecustomized”. Please, correct them.
4- At line 49 you use for the first time CAD/CAM abbreviations without their complete name before, please add them.
5- Other missing spaces at lines 51, 53 and 66. Please, correct them.
Material and Method
Please replace with Materials and Methods.
Study design
1- This section is full of missing spaces and uppercase letters instead of lowercase. An accurate check is requested.
2- Please, add an additional table reporting inclusion/exclusion criteria.
3- Why is the age cut-off at 75 years old? Please, provide more information to justify this threshold for exclusion.
4- Although in the manuscript is clearly reported the primary outcome (such as in the “Evaluation section”), secondary outcomes are not well-defined. In addition, a sentence explaining why you have chosen outcome measures (eg. widely used in the literature, etc...) and how would they fit your primary and secondary outcome measures, is needed. In my opinion, the “Study design” section may be the right paragraph to include these information.
Surgical procedure
This section is very detailed regarding the implant fixation procedure, but no information was provided about patient management (e.g., anesthesia regimen, drugs for pain relief, length of stay, post-operative recovery, etc…).
Please, improve this section.
Other points to be addressed
1- Please, better define your clinical trial. Throughout the manuscript, your study is presented as: a) a prospective, single-arm, single-center, investigator-initiated trial; b) an investigator-initiated clinical trial. Are you sure it is the correct terminology for this kind of study?
2- This reviewer is not clear whether all of the patients enrolled into trial were properly accounted for in the interpretation of results, as there is no description in the methodology towards attempts to address this issue for instance with an intention to treat analyses.
3- Was there a medical monitor or DSMB to ensure patient safety throughout the trial?
4- Were there any planned interim analyses? What was the stopping rules for harm and for futility? Please specify.
5- Since Authors assert that “However, donor site morbidity, prolonged hospitalization time, and a lengthy recovery period with delayed functional rehabilitation are frequently noted [2]. Hence, there is an increasing demand for alternatives to microvascular free osteocutaneous flaps” (lines 40 – 43), what are the advantages of the presented procedure in terms of length of stay and costs reduction? Please, provide these information in the Discussion section.
Author Response
This study, presented as a clinical trial, aims to evaluate the clinical efficacy and safety of specific bioactive glass-ceramic implants for reconstructing zygomatic bone defects. Although this is an area of considerable interest and this study has some strengths, there are several points that need clarification and improvement. In addition, a substantial English language revision and editing work are strongly recommended.
: Thank you very much for the review of the manuscript. We have now carried out substantial revision according to the your comments, which the details of the corrections are summarized below.
We are very grateful for the review of the manuscript and comments that improved the quality of the manuscript, and hope that the corrections would be adequate for this articles to be accepted by your journal.
Abstract
1- In the abstract is missing the number of participants enrolled in this trial. Please, add this information.
We appreciate your comments. We amended the abstract, results as follows;
A total of 8 patients were enrolled in the study. Two patients underwent simultaneous reconstruction of the left and right malar defects using a BGS-7 3Dprinting implant
2- At line 21, it is not clear if “other variables” are considered as secondary outcomes. Please, specify this sentence.
Thank you for your comment and sorry about the confusion. The corresponding sentence is now deleted and the abstract was revised with addition of sentence in the abstract as below;
The secondary outcomes including implant immobilization, satisfaction assessment, osteolysis, subsidence of the BGS-7 implant, and safety were assessed
Introduction
1- At line 36, please replace “its” with “their”.
I changed it as you recommended
facial shape to their original form
2- Introduction should be slightly improved and elaborated when you assert that “there are many options for the reconstruction of maxillofacial bone defects” (lines 38 - 40).
Please, add some sentences regarding the use of bone allografts in oro-maxillofacial surgical interventions and references such as “Barone, A.; Covani, U. Maxillary alveolar ridge reconstruction with nonvascularized autogenous block bone: clinical results. J Oral Maxillofac Surg 2007 ,65, 2039-46” and “Grassi, F.R.; Grassi, R.; Vivarelli, L.; Dallari, D.; Govoni, M.; Nardi, G.M.; Kalemaj, Z.; Ballini, A. Design Techniques to Optimize the Scaffold Performance: Freeze-dried Bone Custom-made Allografts for Maxillary Alveolar Horizontal Ridge Augmentation. Materials 2020, 13, 1393” may be enclosed.
We appreciate your comments. We added the references as you recommended as follows;
Grafting with allograft bone has been documented to be a useful tool in reconstructing jaw bone defect [2,3].
- Barone, A.; Covani, U. Maxillary alveolar ridge reconstruction with nonvascularized autogenous block bone: clinical results. J Oral Maxillofac Surg 2007 ,65, 2039-46
- Grassi, F.R.; Grassi, R.; Vivarelli, L.; Dallari, D.; Govoni, M.; Nardi, G.M.; Kalemaj, Z.; Ballini, A. Design Techniques to Optimize the Scaffold Performance: Freeze-dried Bone Custom-made Allografts for Maxillary Alveolar Horizontal Ridge Augmentation. Materials 2020, 13, 1393
3- At line 47 there are two missing spaces: “by3D” and “arecustomized”. Please, correct them.
We corrected that as follow
by 3D printing technology are customized
4- At line 49 you use for the first time CAD/CAM abbreviations without their complete name before, please add them.
We corrected that as follow
CAD/CAM (computer aided design/computer aided manufacturing)
5- Other missing spaces at lines 51, 53 and 66. Please, correct them.
We corrected those.
Material and Method
Please replace with Materials and Methods.
We replaced with “Materials and Methods”.
Study design
1- This section is full of missing spaces and uppercase letters instead of lowercase. An accurate check is requested.
:Thank you very much for your comments. Extensive editing of the manuscript now have been carried
2- Please, add an additional table reporting inclusion/exclusion criteria.
: Thank you for your comment. We now have added the additional table reporting inclusion/exclusion criteria in the Materials and methods as below
|
Inclusion criteria |
Exclusion criteria |
|
1.Adults aged 19 to under 75 who have completed craniofacial growth 2. Those with obvious zygoma defects, who need and want reconstruction 3. In the case of women of childbearing potential, those who consent to contraceptive use during the clinical trial participation period (more than 6 months after surgery) 4. Those who voluntarily agreed to participate in clinical trials and were willing to comply with the study protocol.
|
1. People with uncontrolled metabolic diseases (e.g. diabetes, osteomalacia, thyroid disease) 2. People who are taking or are planning to take drugs that can affect bone metabolism (Bisphosphonate, Recombinant human parathyroid hormone, Denosumab, etc.) 3.Those with uncontrolled gingivitis, periodontitis and dental caries 4. People with severe heart disease or severe liver dysfunction 5. Persons with infectious diseases with a risk of recurrence 6. People with blood diseases (leukemia, hemophilia, sepsis, etc.) 7. People who cannot stop taking steroids, anti-thrombotics, or anticoagulants before surgery 8. People who cannot stop systemic corticosteroids or anabolic steroids for 3 months after surgery 9. People with osteomalacia and Paget’s disease 10. People who have experienced radiation therapy at the surgical site 11. People who are allergic to implant materials 12. Patients with syphilis and severe epilepsy 13. Drug abuse or alcoholics 14. Smokers 15. People who are pregnant or have a pregnancy plan during the clinical trial period 16. In case the investigator judges that participation in the clinical trial is inappropriate because other, ethical or clinical trial results may be affected. |
3- Why is the age cut-off at 75 years old? Please, provide more information to justify this threshold for exclusion.
: According to the general anesthesia protocol of the hospital where this clinical trial was conducted, patients over 75 years of age must perform echocardiography and lung function tests to receive general anesthesia. Because of the cost of additional tests, more than 75 patients were excluded from this clinical trial.
4- Although in the manuscript is clearly reported the primary outcome (such as in the “Evaluation section”), secondary outcomes are not well-defined. In addition, a sentence explaining why you have chosen outcome measures (eg. widely used in the literature, etc...) and how would they fit your primary and secondary outcome measures, is needed. In my opinion, the “Study design” section may be the right paragraph to include these information.
: Thank you very much for your comments, and we also agree that such information would be important. We now have added some of detailed explanations to study design section as highlighted text below;
The primary outcome variable was the fusion between the implant and the bone evaluated by computed tomography (CT) at 6months after surgery. The secondary variables including immobilization of the implant by CT, a visual analog scale (VAS) for satisfaction, osteolysis, subsidence of the BGS-7 implant, and safety were assessed. A 4-year follow-up study evaluating fusion rates and clinical outcomes using BGS-7 spine cage in 1-level posterior lumbar interbody fusion has been performed [9]. Since the spine study used the BGS-7 spine cage which is same material used in this study, the primary outcome and secondary outcomes used in that study were also evaluated in this study.
Surgical procedure
This section is very detailed regarding the implant fixation procedure, but no information was provided about patient management (e.g., anesthesia regimen, drugs for pain relief, length of stay, post-operative recovery, etc…).
Please, improve this section.
: Thank you very much for your comments, and we also agree that such information would be important. We now have added some of detailed explanations to surgical procedure section as highlighted text below;
All patients received the same anesthetic protocol. Aesthesia was induced with intravenous administration of 2 mg/kg propofol and 0.6 mg/kg rocuronium. Anesthesia was maintained using 2–3% sevoflurane in 1.5 L/min N2O and 1.5 L/min O2. Noninvasive arterial blood pressure, electrocardiography, and pulse oximetry were monitored continuously. Lactated Ringer’s solution (3–6 mL/kg/h) and 6% hydroxyethyl starch were infused throughout surgery. Ketorolac of 30 mg was administered just before the end of surgery and 5 mL of dexamethasone was administered intravenously after surgery.
Other points to be addressed
1- Please, better define your clinical trial. Throughout the manuscript, your study is presented as: a) a prospective, single-arm, single-center, investigator-initiated trial; b) an investigator-initiated clinical trial. Are you sure it is the correct terminology for this kind of study?
: Thank you very much for your comments and sorry about the confusion and vagueness in terms. I changed the term as “ a prospective, single-arm, single-center, clinical trial”
2- This reviewer is not clear whether all of the patients enrolled into trial were properly accounted for in the interpretation of results, as there is no description in the methodology towards attempts to address this issue for instance with an intention to treat analyses.
: Thank you very much for your comments and sorry about the confusion as the writings on the manuscript was not what we exactly intended.
As I mentioned on Evaluation section, the evaluation of CT used for bone fusion (primary outcome) was performed by an independent and experienced oral and maxillofacial surgeon who did not participate in the clinical trial.
3- Was there a medical monitor or DSMB to ensure patient safety throughout the trial?
: Monitoring was performed regularly while the researcher was conducting the research. Monitoring of compliance with the clinical trial plan, maintenance of appropriate and accurate records, consent of subjects, reports of adverse reactions, and compliance with GCP regulations were conducted regularly.
4- Were there any planned interim analyses? What was the stopping rules for harm and for futility? Please specify.
: In the case of clinical trials using 3D printed implants, regular due monitoring must be performed by the Korea Food and Drug Administration. During this clinical trial, the Korea Food and Drug Administration visited the Chung-Ang University Hospital and conducted monitoring, and we received a review result indicating that there was no problem with the ongoing clinical trial. If an adverse reaction occurs, the clinical trial should be stopped.
5- Since Authors assert that “However, donor site morbidity, prolonged hospitalization time, and a lengthy recovery period with delayed functional rehabilitation are frequently noted [2]. Hence, there is an increasing demand for alternatives to microvascular free osteocutaneous flaps” (lines 40 – 43), what are the advantages of the presented procedure in terms of length of stay and costs reduction? Please, provide these information in the Discussion section.
: Thank you for your comment and we also agree that such information would be important. Now, the highlighted text was added in the Discussion section
When performing zygoma reconstruction surgery using a patient specific BGS-7 implant, most of the surgery took less than an hour because there was no need to harvest bones elsewhere in the body. Patients were discharged the day after surgery because there was no donor site morbidity and it took less time for surgery. As hospitalization period was shortened, overall medical expenses also decreased. (data was not shown)
Once again, thank you very much for all of valuable comments made by editors and reviewers.
Sincerely yours,
Reviewer 2 Report
This manuscript describes a clinical trial where additively manufactured glass ceramic BGS-7 implants were used to treat zygomatic defects. The results presented are a 6 month follow-up after implantation. CT results showed that the bone and implants had fused by 6 months post implantation and the implants showed very little displacement from the implanted location. The implants, being fabricated using additive manufacturing were patient-specific, which is advantageous for these types of applications. Comments: Minor: 1. Proof reading for grammar and space errors: page 2 of 12, line 47, space needed between “by” and “3D”, and “are” and “customized” Page 3, line 82, space between “before” and “participating” Page 3 line 84, space between “with” and “obvious” and “zygoma” Page 3 line 94, space between “people” and “who” There are more after that, but close proofreading should be completed. 2. I think the statement in the abstract and on page 8, line 225 “Cone beam CT analysis showed that bone fusion at 6 months after surgery was 100% (10/10)” should be clarified. It makes it seem as though the fusion is 100% in all patients, whereas later it says that fusion is above 50%. I would reword to say that all patients exhibited fusion instead of fusion was 100%. 3. What is the y-axis units for figure 5? 4. What is the reference colors for figure 6? (Like a scale bar?) and can you put a tiny bit of white space between the images? They begin to blend together.Author Response
This manuscript describes a clinical trial where additively manufactured glass ceramic BGS-7 implants were used to treat zygomatic defects. The results presented are a 6 month follow-up after implantation. CT results showed that the bone and implants had fused by 6 months post implantation and the implants showed very little displacement from the implanted location. The implants, being fabricated using additive manufacturing were patient-specific, which is advantageous for these types of applications.
Comments: Minor:
- Proof reading for grammar and space errors: page 2 of 12, line 47, space needed between “by” and “3D”, and “are” and “customized” Page 3, line 82, space between “before” and “participating” Page 3 line 84, space between “with” and “obvious” and “zygoma” Page 3 line 94, space between “people” and “who” There are more after that, but close proofreading should be completed.
: Thank you very much for your comments. Extensive editing of the manuscript now have been carried out
- I think the statement in the abstract and on page 8, line 225 “Cone beam CT analysis showed that bone fusion at 6 months after surgery was 100% (10/10)” should be clarified. It makes it seem as though the fusion is 100% in all patients, whereas later it says that fusion is above 50%. I would reword to say that all patients exhibited fusion instead of fusion was 100%.
: Thank you for your comment and sorry about the confusion. The corresponding sentence is now changes as recommended as below;
Cone beam CT analysis showed that all patients exhibited bone fusion at 6 months after surgery (10/10).
- What is the y-axis units for figure 5?
: 3D printing BGS-7 implant's long axis length is x, minor axis length is y, and thickness is Z.
- What is the reference colors for figure 6? (Like a scale bar?) and can you put a tiny bit of white space between the images? They begin to blend together.
: Thank you so much for your comment. I amended that figure 6. As you recommended.
Reviewer 3 Report
I'm very glad to become acquainted with such interesting research. Hope that my review will help to improve the quality of the article.
Some words in the text are glued. Ex. line 47 “Since the implants developed by3D printing”. The text should be revised and spaces should be added.
In the text, there are no links in Figure 1A, 1D, 2D
In the text for figure citing used "Figure" and "Fig".
Screenshots on figure 6 should be split by black line otherwise it’s hard to distinguish them.
Paragraph 3.2 should be checked and revised. E.g., it’s not clear about what "cases" are talking inline 229. Is it a significant difference between all cases or some part of cases?
A plot legend should be added in figure 6.
There are only 15 items in "References", but in text, there are links on items 16-18. The references list should be checked. And, in my opinion, the reference list can be expanded.
I advise authors to do a literature review in CT-based bone analysis methods, e.g. works of such authors like P. Zysset, T. Grupp, P. Marcian, O. Sachenkov, and etc. And, maybe, use some of the methods in the future to expand the research.
Author Response
Some words in the text are glued. Ex. line 47 “Since the implants developed by3D printing”. The text should be revised and spaces should be added.
:Thank you very much for your comments. Extensive editing of the manuscript now have been carried out
In the text, there are no links in Figure 1A, 1D, 2D
: Thank you for your comment and sorry about the confusion. I put the 1A, 1D, and 2D as follows,
The CBCT data of the maxillofacial regions were obtained with a 0.4 mm voxel size and 512 × 512 matrices using 120 kVp, 11 mA, 17.8 seconds of scan time, and a 12-inch detector field (Figure. 1A.)
. However, since there was a possibility that the implant could break during the screw tightening process, the plate was fixed by pressing the implant downward (figure. 1D).
The multi-planar plane is re-oriented so that the longest axis of the implant is perpendicular to the reference plane, and the minor axis is horizontal to the reference plane (Figure. 2B, C, D.).
In the text for figure citing used "Figure" and "Fig".
: Thank you for your comment and sorry about the confusion. I changed all "fig" to "figure".
Screenshots on figure 6 should be split by black line otherwise it’s hard to distinguish them.
: Thank you so much for your comment. I amended that figure 6. As you recommended.
Paragraph 3.2 should be checked and revised. E.g., it’s not clear about what "cases" are talking inline 229. Is it a significant difference between all cases or some part of cases?
Thank you for your comment and sorry about the confusion. I corrected that corresponding sentence as follow.
The LMEM showed no evidence of significant differences in average fusion rate between each cases. (F[9, 9] = 0.900, P = 0.561),
A plot legend should be added in figure 6.
: Thank you so much for your comment. I amended that figure 6. As you recommended.
There are only 15 items in "References", but in text, there are links on items 16-18. The references list should be checked. And, in my opinion, the reference list can be expanded.
Thank you for your comment and sorry about the confusion. I added the missing references and added 2 more references
Barone, A.; Covani, U. Maxillary alveolar ridge reconstruction with nonvascularized autogenous block bone: clinical results. J Oral Maxillofac Surg 2007 ,65, 2039-46
Grassi, F.R.; Grassi, R.; Vivarelli, L.; Dallari, D.; Govoni, M.; Nardi, G.M.; Kalemaj, Z.; Ballini, A. Design Techniques to Optimize the Scaffold Performance: Freeze-dried Bone Custom-made Allografts for Maxillary Alveolar Horizontal Ridge Augmentation. Materials 2020, 13, 1393
Cho, J.; Kwon, J. S.; Lee, U. L., Occlusion-Fit Three-Dimensional-Printed Zygoma Repositioner. The Journal of craniofacial surgery 2018, 29, (3), 731-732.
Baek, R. M.; Kim, J.; Lee, S. W., Revision reduction malarplasty with coronal approach. Journal of plastic, reconstructive & aesthetic surgery : JPRAS 2010, 63, (12), 2018-24.
Lee, J. H.; Ryu, H. S.; Seo, J. H.; Lee, D. Y.; Chang, B. S.; Lee, C. K., Negative effect of rapidly resorbing properties of bioactive glass-ceramics as bone graft substitute in a rabbit lumbar fusion model. Clinics in orthopedic surgery 2014, 6, (1), 87-95.
Lee, J. H.; Ryu, H. S.; Lee, D. S.; Hong, K. S.; Chang, B. S.; Lee, C. K., Biomechanical and histomorphometric study on the bone-screw interface of bioactive ceramic-coated titanium screws. Biomaterials 2005, 26, (16), 3249-57.
I advise authors to do a literature review in CT-based bone analysis methods, e.g. works of such authors like P. Zysset, T. Grupp, P. Marcian, O. Sachenkov, and etc. And, maybe, use some of the methods in the future to expand the research.
:
Thank you for your comment and we also agree that references would be important. I searched the papers of authors like P. Zysset, T. Grupp, P. Marcian, and O. Sachenkov that you provided. I couldn't find any of the CT analyzes associated with this paper because of my lack of search ability. I will definitely refer to it when doing research on CT analysis in the future. Thank you so much for the good information.
We are very grateful for the review of the manuscript and comments that improved the quality of the manuscript, and hope that the corrections would be adequate for this articles to be accepted by your journal.
Reviewer 4 Report
Dear sir,
Thank you for the opportunity to peer – review this manuscript.
Introduction
Please cite every sentence in the introduction. Do not leave any sentence uncited even if they are from the same citation.
Material and Method
Line 81 – please cite the Declaration of Helsinki as of 2000. This is mandatory.
Discussion
Please cite every sentence that is not deducted directly from this study.
This study is interesting and should be taken into consideration for publication.
Regards,
Author Response
Introduction
Please cite every sentence in the introduction. Do not leave any sentence uncited even if they are from the same citation.
: Thank you for your comment and sorry about the confusion. I added the missing references
Barone, A.; Covani, U. Maxillary alveolar ridge reconstruction with nonvascularized autogenous block bone: clinical results. J Oral Maxillofac Surg 2007 ,65, 2039-46
Grassi, F.R.; Grassi, R.; Vivarelli, L.; Dallari, D.; Govoni, M.; Nardi, G.M.; Kalemaj, Z.; Ballini, A. Design Techniques to Optimize the Scaffold Performance: Freeze-dried Bone Custom-made Allografts for Maxillary Alveolar Horizontal Ridge Augmentation. Materials 2020, 13, 1393
Material and Method
Line 81 – please cite the Declaration of Helsinki as of 2000. This is mandatory.
Thank you for your comment and sorry about the missing. I added following reference and cite that.
World Medical Association General, A., World Medical Association Declaration of Helsinki: ethical principles for medical research involving human subjects (revised October 7, 2000). HIV clinical trials 2001, 2, (1), 92-5.
Discussion
Please cite every sentence that is not deducted directly from this study.
Thank you for your comment and sorry about the confusion. I added the missing references
Cho, J.; Kwon, J. S.; Lee, U. L., Occlusion-Fit Three-Dimensional-Printed Zygoma Repositioner. The Journal of craniofacial surgery 2018, 29, (3), 731-732.
Baek, R. M.; Kim, J.; Lee, S. W., Revision reduction malarplasty with coronal approach. Journal of plastic, reconstructive & aesthetic surgery : JPRAS 2010, 63, (12), 2018-24.
Lee, J. H.; Ryu, H. S.; Seo, J. H.; Lee, D. Y.; Chang, B. S.; Lee, C. K., Negative effect of rapidly resorbing properties of bioactive glass-ceramics as bone graft substitute in a rabbit lumbar fusion model. Clinics in orthopedic surgery 2014, 6, (1), 87-95.
Lee, J. H.; Ryu, H. S.; Lee, D. S.; Hong, K. S.; Chang, B. S.; Lee, C. K., Biomechanical and histomorphometric study on the bone-screw interface of bioactive ceramic-coated titanium screws. Biomaterials 2005, 26, (16), 3249-57.
This study is interesting and should be taken into consideration for publication.
: Thank you very much for your encouragement.
Once again, thank you very much for all of valuable comments made by editors and reviewers.
Sincerely yours,